# A Real-Time License Plate Detection and Recognition Model in Unconstrained Scenarios

**DOI:** 10.3390/s24092791

**Published:** 2024-04-27

**Authors:** Lingbing Tao, Shunhe Hong, Yongxing Lin, Yangbing Chen, Pingan He, Zhixin Tie

**Affiliations:** 1School of Computer Science and Technology, Zhejiang Sci-Tech University, Hangzhou 310018, China; lb_tao@zstu.edu.cn (L.T.); 202020502017@mails.zstu.edu.cn (S.H.); eluian@zstu.edu.cn (Y.L.); cyb@zstu.edu.cn (Y.C.); 2Keyi College, Zhejiang Sci-Tech University, Shaoxing 312369, China; 3School of Science, Zhejiang Sci-Tech University, Hangzhou 310018, China; pinganhe@zstu.edu.cn

**Keywords:** license plate recognition, multi-head attention, global feature extractor network, parallel decoder, YOLOv5

## Abstract

Accurate and fast recognition of vehicle license plates from natural scene images is a crucial and challenging task. Existing methods can recognize license plates in simple scenarios, but their performance degrades significantly in complex environments. A novel license plate detection and recognition model YOLOv5-PDLPR is proposed, which employs YOLOv5 target detection algorithm in the license plate detection part and uses the PDLPR algorithm proposed in this paper in the license plate recognition part. The PDLPR algorithm is mainly designed as follows: (1) A Multi-Head Attention mechanism is used to accurately recognize individual characters. (2) A global feature extractor network is designed to improve the completeness of the network for feature extraction. (3) The latest parallel decoder architecture is adopted to improve the inference efficiency. The experimental results show that the proposed algorithm has better accuracy and speed than the comparison algorithms, can achieve real-time recognition, and has high efficiency and robustness in complex scenes.

## 1. Introduction

The license plate of a vehicle is a crucial identifier, for which accurate and real-time recognition has a very wide range of applications, such as vehicle identification, intelligent toll collection, vehicle density statistics, access control management, intelligent driving, traffic control, and traffic scene understanding. In recent years, license plate recognition has received broad attention from a wide range of researchers [1,2].

License plate detection and recognition methods can be roughly classified into traditional methods and deep learning based methods. Most traditional methods use manually selected features to locate the position of license plates [3,4,5], then morphological methods are used to separate the characters in license plates before recognizing individual characters [6,7]. However, the accuracy of these methods is not high, especially in complex scenes. Deep learning-based methods generally contain two networks: a detection network and recognition network. The detection network is used to locate the location of license plate in the picture, and the recognition network is used to recognize the sequence of characters in the license plate picture.

Deep learning-based license plate detection algorithms are mainly classified into two-stage detection algorithms [8,9,10] and one-stage detection algorithms [11,12,13,14]. The two-stage detection algorithms have better performance; however, the real-time performance is much inferior to the one-stage detection algorithms. The single-stage detection algorithms are more suitable for license plate detection in a variety of application scenarios and can meet the requirements of both high detection accuracy and real-time performance.

Depending on whether or not segmentation of character positions in license plate images is required, deep learning-based license plate recognition methods can be divided into two categories: methods that require segmentation and methods that do not require segmentation. Methods that require segmentation [15,16] have high recognition accuracy but slow recognition speeds and insufficient generalization performance. Methods that do not require segmentation can be categorized into three subcategories. The first subcategory requires correction of the license plate before recognition [17,18,19], resulting in higher recognition accuracy; however, these methods are calculation-intensive and slow inference. The second subcategory uses a Recurrent Neural Network (RNN) combined with an attention mechanism for license plate recognition [20,21,22,23,24]; however, they cannot parallelize the operation and are not efficient. The third subcategory is to share license plate features between the detection network and the recognition network [25,26], which improves the efficiency of feature vector utilization and reduces the computational effort; however, the recognition accuracy is insufficient, and the anti-interference ability is weak and the robustness is poor.

Despite considerable progress in LPDR research, existing algorithms are mainly designed for recognition in stationary and constrained scenarios. License plate recognition in natural environments faces multiple challenges, including variations in shooting distance, tilt angle, light intensity, weather conditions, and image blurring [1,27]. These factors increase the complexity of accurately recognizing license plates in natural scenes and pose a significant scientific challenge to researchers in the field.

Recently, Transformer has made significant achievements in the field of Natural Language Processing (NLP) [28], and it is gradually being used in the field of computer vision with outstanding achievements [29,30]. Jocher [31] improved the YOLO algorithm to achieve high accuracy and speed results on the target detection task. In this paper, inspired by the above works and in order to be able to recognize license plates accurately and in real time, Transformer is applied to the license plate recognition task and a novel license plate detection and recognition model named YOLOv5-PDLPR is proposed. The model uses the YOLOv5 target detection algorithm in the license plate detection part, which can improve the inference speed and accuracy of the entire system. The parallel decode license plate recognition algorithm (PDLPR) is proposed to recognize license plates in the license plate recognition part. Its feature extraction network introduces the Focus Structure, and the decoding structure uses a parallelized decoder to replace the mainstream serial RNN. Through the Multi-Head Attention mechanism, PDLPR is able to comprehensively extract the feature information of the image, which can precisely locate the character in the license plate image. Its input license plate picture can be recognized without correction and segmentation to obtain the right result, which improves the accuracy of license plate recognition comprehensively. Meanwhile, the parallelized structure design of the decoder can increase the inference speed and save the inference time. The main contributions of this paper are as follows:We propose a new license plate detection and recognition model called YOLOv5-PDLPR, which employs the YOLOv5 target detection algorithm in the license plate detection part and the newly proposed license plate recognition algorithm PDLPR. PDLPR has three main newly designed components: a Multi-Head Attention mechanism for accurately recognizing individual characters, a feature extraction network for improving the integrity of the global feature extraction network, and a state-of-the-art parallel decoder architecture for improving inference efficiency.Experimental results on the CCPD dataset [25] show that the proposed method achieves an average accuracy of 99.4% and a recognition speed of 159.8 FPS, which are better than those of the comparison algorithms.

The rest of the paper is organized as follows: The techniques relevant to license plate detection and recognition are introduced in Section 2. The proposed license plate recognition algorithm YOLOv5-PDLPR is introduced in Section 3. Section 4 describes the datasets used in the experiments, the essential parameter settings of the neural network, and the evaluation metrics related with the experiment results. Section 5 gives the experimental results and comparative analysis. Section 6 performs an ablation study to verify the efficacy of the proposed method. Finally, conclusions are drawn in Section 7.

## 2. Related Work

### 2.1. License Plate Detection

License plate detection is the foundation of license plate recognition, and its accuracy directly affects the results of character recognition in license plates. Currently, there are two main categories of license plate detection methods in common use. One is the license plate detection algorithm based on traditional methods. The other is the deep learning-based license plate detection algorithms.

License plate detection algorithms based on traditional methods usually extract the intrinsic properties of license plates such as edges, colors, local textures, and morphological analysis as manual image features for license plate detection [32], such as edge feature-based license plate detection algorithms [33,34], color feature-based license plate detection algorithms [35,36], texture feature-based license plate detection algorithms [37,38], character feature-based license plate detection algorithms [39,40], and detection algorithms based on more than two features [35]. As noise interference occurs in real license plate images, most methods based on a single manual feature only work in specific scenes and have poor detection results.

Given sufficient training data, deep learning based license plate detection algorithms have powerful feature representation and high performance compared to license plate detection algorithms based on traditional methods. They can usually be divided into two categories: one-stage detection methods and two-stage detection methods.

Fast R-CNN [8] and Faster R-CNN [9] are classical two-stage detection networks that employ the region proposal network in the first stage to share the convolutional features of the whole image and generate high-quality region proposal candidate frames. Then, in the second stage, the Convolution Neural Network (CNN) classifier is used to classify the candidate frames and obtain the kind of targets.

Although two-stage license plate detection methods detect objects accurately and quickly, they are slow and cannot meet the needs of real-time detection tasks. Therefore, one-stage detection methods have emerged, and the representative networks include the YOLO series [11,12,13,14], TE2E [41], SSD [42], CA-CenterNet [43], YOLOv5 [31], Optical Flow CNN algorithms [44,45], etc.

### 2.2. License Plate Recognition

After a license plate positioning is complete, character recognition of the license plate is a significant task, whose task is to recognize the sequence of the license plate characters in the license plate image. Traditional license plate character recognition methods and deep learning license plate character recognition methods are the two primary categories of methods.

#### 2.2.1. Traditional License Plate Character Recognition Method

Traditional license plate character recognition methods have two main steps: license plate character segmentation and character recognition. Segmentation of license plate characters is to separate all characters of a license plate before character recognition in order to match the input of the character recognition algorithm. The usual character segmentation methods are character detection, concatenated domain search, and vertical projection. The segmented individual characters are then fed into the recognition module to obtain the recognition results. Character recognition methods usually include template matching, feature statistics, and machine learning.

Maglad et al. [46] segmented characters using the Connected Component Analysis (CCA) method, determining different connected domains to slice the characters. Hsu et al. [47] segmented the characters within a license plate picture using the Maximally Stable Extremal Regions (MSER) technique. Rahman et al. [48] used vertical projection to find the location of the start and end of the characters within a license plate image, used horizontal projection to obtain individual characters, and then used template matching to recognize the characters. Gou et al. [6] distinguished different license plate characters with the help of limit regions of specific characters, and then character recognition was performed using Restricted Boltzmann Machines (RBMs). Ashtari et al. [7] employed an improved template matching-based method to locate the license plates. A hybrid classifier consisting of a decision tree and a support vector machine (SVM) with a homogeneous fifth degree polynomial kernel is then applied to identify the extracted letters and numbers.

#### 2.2.2. Deep Learning-Based License Plate Character Recognition Method

Traditional methods are very susceptible to the influence of uneven illumination and noise in the license plate image, which leads to the wrong recognition of some license plate characters and low accuracy of license plate sequence recognition. Deep learning-based methods for license plate recognition have shown excellent accuracy and speed in practical applications, and become the current research hotspots.

(1)Methods that require character segmentation

Bjorklund et al. [49] proposed a license plate recognition system based on convolutional neural networks and designed two CNN networks, one for license plate detection and the other for character detection and classification. Yao et al. [50] presented a novel CNN for license plate recognition and improved the learning ability of the network by learning image features from coarse to fine. Zhuang et al. [15] introduced an approach to license plate recognition based on semantic segmentation, which converts the license plate recognition problem into a semantic segmentation task and generates recognition results using the semantic type of each pixel. However, the subsequent processing steps introduced by this method, such as connected component processing and character counting, result in poor generalization performance for the algorithm and require careful adjustment. Castro et al. [16] put forward an improved SSD network to recognize license plate characters. It locates the region of the license plate, as well as the area of the characters, and then segments the characters for recognition, and the recognition accuracy of this method is higher.

Although such methods improve recognition accuracy, this is achieved by laborious data annotation, i.e., annotation of each character in the license plate image, increasing the cost of data annotation.

(2)Methods without character segmentation

Recently, several approaches have attempted to jointly extract semantic and location information that can achieve better recognition performance without segmenting the characters in license plate images. These methods can be broadly classified into three categories as follows.

The first category is to feed license plate images or feature maps into a correction network for correction before the license plate images is recognized. The corrected license plate images or feature maps are then fed to the license plate recognition network for recognition. The LPRNet proposed by Zherzdev et al. [17] used a Space Transformation Network (STN) [51] to affine transform the license plate image to correct the shape of the license plate, which makes the license plate image better recognized. Zhang et al. [18] proposed the Inverse Compositional Spatial Transformer Network (ICSTN), which combines Thin Plate Spline (TPS) and STN for correcting license plate images with geometric deformation to improve the accuracy of license plate recognition. Yousaf et al. [19] adopted the Multi-Object Rectification Network (MORN) [52], which chunks the image and then adjusts the offset of each block to correct the license plate image before recognition. This method has been successful in the recognition of Pakistani license plates. This category of methods can correct license plates with skewed angles and improve recognition accuracy. However, it increases the computational burden of the training process and decreases the inference speed due to the addition of a correction network.

The second category is to convert the license plate task into sequence recognition task using RNN. Zou et al. [23] introduced BiLSTM to locate each character of a license plate by heat map visualization without segmenting each character. Moreover, they utilized 1D attention to extract features of character regions. Zhang et al. [24] proposed a license plate character recognition network with an Xception network for feature extraction and a RNN decoder combined with a 2D attention mechanism. In real-world circumstances, it can recognize license plates with both regular and irregular patterns. Gao et al. [20] proposed a novel license plate recognition method using a two-stage encoder combined with a Long Short Term Memory (LSTM) decoder. The method is able to improve the coding quality and can recognize various types of license plates. This category of methods uses RNN networks that treat license plates as a sequence recognition problem; however, there is a large number of cyclic computations in RNN networks that cannot be computed in parallel, increasing the inference time. Moreover, the long-term dependency of LSTM leads to performance degradation.

The third category is to use the features extracted by the license plate detection network for license plate recognition. The method proposed by Xu et al. [22] first performed a region-of-interest (ROI) pooling operation on the feature map generated by the detection part to obtain the feature vectors of the license plate regions. The features of the license plate are then sent into the classifier in the recognition portion to acquire the license plate sequence. The approaches proposed by Gong et al. [21] and Qin et al. [26] used the Feature Pyramid Networks (FPN) in the license plate detection part to extract the shared features for classification and recognition. The detection branch then generates bounding boxes and corner points, which are used for Region of Interest Align (RoIAlign) and correction, respectively. Finally, the located license plate features are used for recognition to determine the license plate sequence. This category of methods shares the features extracted by the license plate detection network with the license plate recognition network, reducing the calculation cost. However, the recognition networks of this category of methods are designed in a simpler way, and the extracted features are not semantically rich enough, resulting in insufficient recognition accuracy.

### 2.3. Transformer

Vaswani et al. [1] proposed the Transformer architecture, which was initially applied in the fields of machine translation and Natural Language Processing (NLP), as a neural network based mainly on a self-attention mechanism. In contrast to the RNN-based approaches, Transformer makes the training process highly parallel, which can reduce model complexity and improves text recognition accuracy. In recent years, inspired by the successful application of Tranformer in the field of NLP, several works [53,54,55] have proposed the use of Transformer to replace the recursive structure in the seq2seq framework, which facilitates parallel computation and speeds up processing.

Mahdavi et al. [56] used Transformer in the field of mathematical expression recognition. They won the ICDAR 2019 Competition on Recognition of Handwritten Mathematical Expressions and Typeset Formula Detection with the greatest recognition rate. Yang et al. proposed HRGAT [54] for scene text recognition using Transformer. By merging CNN feature maps with a 2D attention map and then linking to the parallel decoder, it can swiftly recognize the text of scenes with irregular spatial distribution. Kang et al. [55] applied Transformer network to a handwritten text recognition task and achieved excellent performance. Ma et al. [57] proposed the Text Attention Network (TATT), which uses CNN and Transformer to align text with spatially distorted text images, achieving state-of-the-art performance in text super-resolution tasks.

## 3. Proposed Method

As shown in Figure 1, the proposed YOLOv5-PDLPR consists of two main parts: the YOLOv5-based license plate detection network and the PDLPR license plate recognition network proposed in this paper. The former receives an entire car picture as its input, locates the license plate position within the picture, and then outputs a picture including only the license plate information. The latter takes the license plate picture as its input and puts the license plate picture through feature extraction, encoding, and decoding operations to obtain the sequence of license plate characters.

### 3.1. License Plate Detection

YOLOv5 was optimized and improved by Glenn et al. [31] on the basis of You Only Look Once (YOLO), which adds mosaic data improvement and adaptive anchor frame calculation at the input side, the Focus Structure and cross stage partial network (CSPNet) in the backbone network, and Generalized Intersection of Union (GIOU) loss in the Prediction part to the YOLO model. These improvements make YOLOv5 more flexible and faster than YOLOv4, with the characteristics of lightness, and has been widely used in the field of target detection in recent years. In this paper, YOLOv5 is used as a license plate detection network.

### 3.2. License Plate Recognition

The proposed license plate recognition model PDLPR is shown in Figure 2, which consists of three main modules: the Improved Global Feature Extractor (IGFE), the Encoder, and the Parallel Decoder. All the input license plate images were initially resized to 48 × 144. In the IGFE module, feature extraction was performed on an image and features were converted into a feature vector of 512 × 6 × 18. In the Encoder module, the position encoder encoded the position of the feature map, added it to the image feature vector, then encoded the vector using Multi-Head Attention to produce a feature vector. In the Parallel Decoder module, the Multi-Head Attention of the decoder was employed to decode the output feature vector from the encoder in order to predict the license plate sequence.

#### 3.2.1. Improved Global Feature Extractor

The Focus Structure was added at the beginning part of the IGEF to implement the feature map downsampling function while ensuring that no feature information was lost. In other parts requiring downsampling, the pooling operation was also replaced with a convolution operation with a stride of two in order to preserve the integrity of the extracted network features. This can improve the accuracy of license plate character recognition. Figure 3 shows the structure of the IGFE module, which consists of a Focus Structure module, two ConvDownSampling modules, and four RESBLOCK modules.

(1)Focus Structure Module

The structure of Focus Structure module is shown in the bottom part of Figure 3, which was used to conduct picture slicing operations, and its operation process is shown in Figure 4, where a value was taken at each interval of one pixel in a input picture so that one picture was equally divided into four feature maps. Then, they were concatenated along the channel direction. Thus a three-channel image became a 12-channel feature map with half the original width and height. Finally, the obtained feature map is convolved to perform the downsampling operation. A Focus Structure is better than other ways of downsampling because it does not lose any feature information. This means that the extracted semantic information will be more comprehensive.

(2)RESBLOCK module

The structure of each RESBLOCK module is shown in the central part of Figure 3, and consisted of two residually connected CNN BLOCK modules. During forward inference, the residual connected structure could prevent the network’s gradient disappearance and explosion.

In the CNN BLOCK module’s convolutional layer for extracting features, we utilized conv2d with stride = 1 and kernelSize = 3 to extract features, which were then passed via the BatchNormalization layer [58] and activation function layer in order to extract the image’s visual features.

The activation function made use of the leakyRelu [59] shown in Figure 5a rather than the Relu [60] shown in Figure 5b. The reason is that when the input of the Relu function is negative, the output is always 0, and its derivative is also 0. This tends to cause dead neurons, which means that the neurons no longer learn and the parameters no longer change. The leakyRelu is given a smaller slope value for the case where the input is negative to avoid the occurrence of death neurons.

(3)ConvDownSampling module

The structure of the ConvDownSampling module in this paper is the same as that of CNN BLOCK. However, we set the stride to 2 in conv2d for downsampling and used the convolution operation in place of the pooling operation for downsampling. This preserves more feature information in the downsampling process, thus improving the accuracy of license plate recognition.

#### 3.2.2. Encoder

As shown in Figure 6, the Encoder in this paper consisted of three encoding units connected by residuals, and each unit contained four submodules: CNN BLOCK1, Multi-Head Attention, CNN BLOCK2, and Add&Norm. The CNN BLOCK1 and CNN BLOCK2 structures are the same as in Section 3.2.1, but with a few differences that will be explained later.

Before calculating Multi-Head Attention, the CNN BLOCK1 in the bottom part of Figure 6 was used to increase the feature vector dimension so that richer feature information could be extracted. Here, we set CNN BLOCK1’s stride to 1, kernelSize to 1, padding to 1, and the dimension of the output to 1024.

The CNN BLOCK1 output feature vectors were then encoded using Multi-Head Attention [1]. Here, parallel processing was used to compute the attention on each subspace. The results in different spatial dimensions are then connected, and a linear conversion is performed to obtain the final encoding result. This can attend to the connections between features in multiple ways and in multiple spaces. The Multi-Head Attention $MHA\left(Q,K,V\right)$ is calculated as shown in Equation (1):(1)$$MHA\left(Q,K,V\right)=\begin{array}{c} {}[{head}_{1}\left({QW}_{1}^{Q},{KW}_{1}^{K},{VW}_{1}^{V}\right),{head}_{2}\left({QW}_{2}^{Q},{KW}_{2}^{K},{VW}_{2}^{V}\right),\ldots ,\\ {head}_{i}\left({QW}_{i}^{Q},{KW}_{i}^{K},{VW}_{i}^{V}\right),\ldots ,{head}_{h}\left({QW}_{h}^{Q},{KW}_{h}^{K},{VW}_{h}^{V}\right)] \end{array}$$
where Q, K, and $V\in R^{n\times d}$, ${head}_{i}\left(QW_{i}^{Q},KW_{i}^{K},VW_{i}^{V}\right)\in R^{n\times d_{k}}$, $W_{i}^{Q}$, $W_{i}^{K}$ and $W_{i}^{V}\in R^{d\times d_{k}}$ $W^{O}\in R^{d\times d}$, n=width∗height=108, d=1024, $d_{k}=\frac{d}{h}=128$, h = 8. ${head}_{i}\left(QW_{i}^{Q},KW_{i}^{K},VW_{i}^{V}\right)$ denotes the result of attention calculation for the i-th subspace; $W_{i}^{Q}$, $W_{i}^{K}$, and $W_{i}^{V}$ are the projection matrices that project Q, K, and V to the i-th subspace, respectively; $W^{O}$ is the matrix for computing the linear conversion of the head; width and height are the width and height of the feature vector output from CNN BLOCK1, respectively. The value of d is equal to the dimensionality of the feature vector output by CNN BLOCK1. h is the number of heads in Multi-Head Attention, which means that the neural network attends to features in h spaces. After the experimental comparison, the license plate recognition accuracy is the highest when h=8, and the experimental results are shown in Section 6. $d_{k}$ is the dimension of the projection vector of the input feature vector on each subspace, which is calculated by dividing d by h. The calculation of Q,K, and V in Equation (1) is shown in Equation (2):(2)$$\left\{\begin{array}{l} X={\left[x_{1},x_{2},\ldots x_{m},\ldots x_{n}\right]}^{T}\\ Q=XW_{Q}\\ K=XW_{K}\\ V=XW_{V} \end{array}\right.$$
where $X\in R^{n\times d}$, $W_{Q}$, $W_{K}$, and $W_{V}\in R^{d\times d}$, $x_{m}\in R^{1\times d}$. X is the feature vector output from CNN BLOCK1. $W_{Q}$, $W_{K}$, and $W_{V}$ are three different trainable weights, which were obtained by random initialization at the very beginning of training and then updated by gradient descent during the training process, and finally the suitable weights are obtained to fit the real values.

Each ${head}_{i}\left(QW_{i}^{Q},KW_{i}^{K},VW_{i}^{V}\right)$ in Equation (1) was calculated as shown in Equation (3):(3)$${head}_{i}\left(QW_{i}^{Q},KW_{i}^{K},VW_{i}^{V}\right)=softmax\left(\frac{QW_{i}^{Q}{\left(KW_{i}^{K}\right)}^{T}}{\sqrt{d_{k}}}\right)VW_{i}^{V}$$
where the softmax function was used to calculate the probability distribution over the feature locations. The use of $\sqrt{d_{k}}$ was to avoid the softmax value tending to 0 when the dot product of $QW_{i}^{Q}$ and ${\left(KW_{i}^{K}\right)}^{T}$ was too large.

After computing Multi-Head Attention, the feature vector dimension is then restored using CNN BLOCK2 in the middle of Figure 6 to ensure the same input and output dimension before and after the encoder unit. Here, we set stride to 1, kernelSize to 1, padding to 1, and the output dimension to 512.

The Add&Norm module connected the feature vectors before and after computing Multi-Head Attention through the residual network, and then performed the Layer Normalization, which can prevent overfitting of the model to a certain extent, as well as speed up the convergence of the model.

#### 3.2.3. Parallel Decoder

The parallel decoder module employs a parallel decoder [1]. In the parallel decoder module, a multi-headed attention mechanism is used to calculate the correlation between the input feature vectors, which matches the feature vector of each character of the license plate, so that only the license plate sequence is required for the training dataset labels, and no individual character positions need to be labeled.

The structure of the parallel decoder is shown in Figure 7, which consists of three decoding units. Each unit contains four sub-modules: Multi-Head Attention, Masked Multi-Head Attention, Feed-Forward Network and Add&Norm. The Multi-Head Attention module and the Add&Norm module were similar to those described in Section 3.2.

The function of the Masked Multi-Head Attention is to prevent the model from focusing on subsequent sequence information and to ensure the parallelism of training. It is implemented by adding the input eigenvector matrix with an upper triangular matrix whose elements are all −∞, and then performing a softmax operation on the summed matrix. This turns the original eigenvector matrix into a lower triangular eigenvector matrix. The masking operation of the Masked Multi-Head Attention is able to restrict the region of attention at each time step, ensuring that the prediction at each location relies only on the known output prior to that location. Due to the design of Masked Multi-Head Attention, the entire training process required a single forward computation. Nevertheless, when the RNN model performs inference, the operation at time t+1 can only continue once the operation at time t has been completed. Therefore, Masked Multi-Head Attention made the inference of the model proposed in this paper significantly faster than the RNN-based model.

The output of the Masked Multi-Head Attention was then fed into the Add&Norm module. This performed a normalization operation to prevent model overfitting and accelerate model convergence.

The CNN BLOCK3 and the CNN BLOCK4 change the dimension of the output feature vector of the Encoder to 512 × 18 before decoding in order to reduce the size of the feature vector and the computational load on the parallel decoder. Then, the outputs of the Add&Norm and Encoder modules are fed into Multi-Head Attention as Q, K, and V, respectively, where K and V contain the feature information of the license plate image, and Q contains the semantic information of the license plate label. The Multi-Head Attention calculates the correlation of each image feature with the labeled text feature. The higher the correlation, the higher the probability that the corresponding location in the image is a certain character. Here in the CNN BLOCK3, we set the convolutional layer parameters stride to 3, kernelSize to (2,1), padding to 1, and output dimension to 512. In the CNN BLOCK4, we set the convolutional layer parameters stride to 3, kernelSize to 1, padding to (0,1), and the output dimension to 512.

The output features of Multi-Head Attention were processed using the Add&Norm module and then input to the Feed-Forward Network module. The Feed-Forward Network module consists of two linear conversions, where the feature vector is input to the first linear function, activated with the ReLU function, and then input to the second linear function. The definition of a Feed Forward Network $FFN\left(\cdot \right)$ is shown in Equation (4):(4)$$FFN\left(x\right)=\mathrm{Max}\left(0,xW_{1}+b_{1}\right)W_{2}+b_{2}$$
where $W_{1}$ and $W_{2}$ are the weights, $b_{1}$ and $b_{2}$ are the biases, and $\mathrm{Max}\left(\cdot ,\cdot \right)$ is the maximum function. $W_{1}\in R^{d\times d}$, $W_{2}\in R^{d\times d}$, $b_{1}\in R^{d}$, and $b_{2}\in R^{d}$. In order to facilitate the connection between model layers, the output size d was set to 512 for all sub-layers in the model.

Finally, the output of the Feed-Forward Network module was employed for forward inference with the Add&Norm module to speed up the convergence of the model.

## 4. Experimental Setup

### 4.1. Datasets

To evaluate the effectiveness of the proposed model YOLOv5-PDLPR, the datasets, as shown in Table 1, were selected.

CCPD [25] is a large and diverse open source dataset of Chinese city license plates, providing 290k images of unique license plates annotated in detail. Each image in this dataset contains only one license plate, and each plate consists of seven characters, of which the first character represents a provincial administrative region (31 categories in total, excluding Taiwan Province, Hong Kong SAR, and Macau SAR), the second character is a letter, and each of the remaining five characters is a letter or a number (all occurrences do not contain “I” and “O”, with 34 categories of numbers and letters). As shown in Table 2, the dataset is grouped into nine sub-datasets according to recognition difficulty, illumination of the license plate area, distance from the plate at the time of capture, horizontal and vertical tilt degree, and weather (rain, snow, fog). During the model comparison experiments, half of the data in the sub-dataset CCPD-base were randomly selected as the training set, while the other half were used as the validation set. Six sub-datasets (CCPD-DB, CCPD-FN, CCPD-Rotate, CCPD-Weather, CCPD-Challenge, and CCPD-Tilt) were selected for testing the models.

PKUData [34], published by Yuan et al., offers pictures of license plates in various situations, but each picture contains only the detection box for the location of the license plate and misses the license plate sequence. Therefore, we manually annotated 2253 images from its data subsets G1 (normal daytime environment), G2 (daytime with sun glare), and G3 (nighttime) to evaluate license plate identification.

CLPD [24] is provided by Zhang et al. It collects 1200 images of different license plates from 31 Chinese provincial administrative regions (excluding Taiwan Province, Hong Kong SAR, and Macau SAR) taken in various environments. Like PKUData, CLPD is only used to test the license plate recognition model.

AOLP [47] consists of 2049 license plate images from the Taiwan Province of China. According to complexity and shooting conditions, the dataset is divided into three subsets: access control (AC), law enforcement (LE), and road patrol (RP). AC contains 681 images, LE contains 757 images, and RP contains 611 images. Each license plate consists of six characters, each consisting of a letter or number (excluding the “O”). As the license plate style of the Taiwan Province is completely different from that of other Chinese provinces, we conducted three sets of experiments on this dataset, each using two of its three subsets for training and the remaining one for testing.

### 4.2. Implementation Details

The experiments were conducted on a server Intel(R) Xeon(R) Silver 4210 CPU @ 2.20 GHz, with a GeForce RTX 2080Ti GPU and the operating system Ubuntu 18.04.6, using the neural network framework PyTorch. To prevent overfitting, we employed data enhancement strategies such as color dithering, random cropping, random scaling, random panning, and random rotation.

The license plate detection algorithm used in this paper is the YOLOv5. The size of the training input image was adjusted to 640 × 640, the Adam optimizer was used to train the detection network for 300 epochs, the batch size was set to 50, the initial learning rate was set to $1\times 10^{-3}$, the final learning rate was $1\times 10^{-5}$, and the cosine variation was used. The license plate detection algorithm was tested with the input image resized to 640 × 640 and the batch size was set to 5.

The training process of the license plate recognition algorithm used in this paper was trained with the input image size adjusted to 48×144, using the Connectionist Temporal Classification loss function and Adam optimizer; the epoch was set to 1000, the batch size was set to 128, and the initial learning rate was set to $1\times 10^{-3}$. When the loss was not reduced, the learning rate was multiplied by a decay factor of 0.9 every 20 epochs. The license plate recognition algorithm was tested with the input image resized to 48 × 144 and the batch size was set to 5.

### 4.3. Evaluation Indicator

The evaluation metric for license plate detection in this paper was Intersection Over Union (IOU), which was defined as shown in Equation (5):(5)$$IOU=\frac{Area(db\cap gb)}{Area(db\cup gb)}$$
where gb is the area of the ground truth box, db is the area of the detected bounding box, Area(·) is a function that finds the area, ∩ is an intersection operation and ∪ is a union operation. For a fair comparison, the same evaluation criteria as [25] were used in this paper on the CCPD dataset. The detected bounding box was considered correct only if the IOU was greater than 0.7. When the IOU was greater than 0.6 and each character on a license plate was correctly recognized, the license plate recognition result was considered correct.

## 5. Experiment Results

The performance of the proposed license plate recognition algorithm YOLOv5-PDLPR and the state-of-the-art approaches were compared in this section using the same experimental setup.

### 5.1. Experiments on the CCPD Dataset

The experimental results of comparing YOLOv5, which was used as the license plate detection algorithm in this paper with the state-of-the-art approaches on the CCPD dataset, are shown in Table 3.

As can be seen from Table 3, the YOLOv5 algorithm has a higher average accuracy on the entire CCPD test dataset than all the comparison algorithms. In addition, the accuracy on the CCPD-DB, CCPD-FN, CCPD-Rotate, CCPD-Tilt, and CCPD-Weather increased by 6.1%, 8.9%, 3.5%, 4.2%, and 17.6%, respectively, when compared to the second-best algorithm. Its detection speed reached 218.3 FPS, which was 155.3% faster than RPnet [24]. Thus, it can be seen that YOLOv5 has the highest detection efficiency.

Table 4 shows the results of a comparison between the proposed framework YOLOv5-PDLPR and the state-of-the-art algorithms on the CCPD dataset.

As seen from Table 4, the average accuracy of the proposed algorithm on the whole CCPD test dataset was 99.4%, and the accuracies on CCPD-base, CCPD-DB, CCPD-FN, CCPD-Weather, and CCPD-Challenge were 99.9%, 99.5%, 99.5%, 99.4%, and 94.1%, respectively, which are higher than all the comparison algorithms. The accuracies obtained using YOLOv5-PDLPR for recognition on the sub-datasets CCPD-Rotate and CCPD-Tilt were 0.1% and 0.3% lower than the method proposed by Fan et al. [43], because they trained their models with synthetic data, which made their models learn more features. However, the recognition results of YOLOv5-PDLPR were better than those of the algorithm of Fan et al. [43], which was not trained using synthetic data. YOLOv5-PDLPR achieved a speed of 159.8 FPS, which was 87.1% faster than the second-fastest algorithm. This is because YOLOv5-PDLPR uses parallel inference to improve efficiency and saves recognition time by not performing additional correction operations after the license plate detection task is completed. Experimental results on this dataset show that our license plate recognition model is robust and efficient in complex scenarios and is a real-time recognition framework that can meet the requirements of road surveillance.

Figure 8 shows the results of detection and recognition of six license plate images by three license plate detection recognition algorithms (Zhang et al.—2020 [24], Xu et al.—2018 [25], YOLOv5-PDLPR (Ours)), respectively. The method of Zhang et al.—2020 [24] does not use the license plate detection network to determine the location of the license plate and then directly inputs the real license plate image for recognition, and the other two methods use the results of the license plate location network for recognition. The sequence of characters after each “GT” in the first row of Figure 8 represents the sequence of real license plate characters, and the visualization result pictures of the method for detecting and recognizing six license plate images are listed after each method name in turn, and the sequence of characters after “Pred” below each visualization detection and recognition picture indicates the sequence of license plate characters detected and recognized by the method, and if a character is in red font, it means that the character is not correctly recognized by the method. As can be seen from Figure 8, some characters are incorrectly recognized by the method of Zhang et al. [24] and Xu et al. [25] in the case of light intensity, a tilted license plate, and a blurred license plate, while our method is able to accurately locate and correctly recognize them, which indicates that our proposed algorithm performs better in complex scenarios.

By plotting the heat map of the model, we can observe where the network is concerned during the run. As shown in Figure 9, there are six columns in total. Each column shows the heat map of different license plate pictures in complex cases. The first row of each column displays the original images of various license plates, while the second row begins with the attention map of each individual character. In each column from top to bottom, if a character on the license plate is darker, it indicates that the network pays more attention to the character features at that location, and then the network extracts these features for recognition. For example, in the second line of the license plate “皖NLE9132”, the color of the characters “皖” is deeper, indicating that the network is more focused on the characteristics of the location of “皖”. Similarly, the network is able to locate the features on other character positions on the license plate and thus accurately identify the characters on the plate.

### 5.2. Experiments on CLPD and PKUData Datasets

After training the models on the CCPD-Base dataset, the models were evaluated for recognition using the CLPD and PKUData datasets, and two cases of recognition are considered: one is to recognize all characters in the license plate (including Chinese characters), and the other is to recognize only non-Chinese characters in the license plate, and is compared with the license plate recognition methods that have performed the same experiments in recent years, and the experimental results are shown in Table 5.

Considering Chinese characters, the test results on the CLPD dataset indicate that the accuracy of the algorithm proposed in this paper is lower than that of the algorithm proposed by Fan [43] using synthetic data. This is because our model does not use synthetic data and is only trained on the CCPD dataset, and the Chinese characters in the CCPD dataset are not evenly distributed, with “皖” accounting for more than 95%. However, when tested on the CLPD dataset, the Chinese characters are evenly distributed, and there were also Chinese characters that do not appear in the CCPD dataset. Consequently, the model was unable to fully learn the Chinese information, resulting in low test accuracy. Nevertheless, the accuracy of our method is higher than the algorithm proposed by Fan [43], which was not trained using synthetic data. When Chinese characters are not considered, our algorithm has the highest accuracy (ACC), and the ACC improves by 5.2% compared to the second-best result.

As shown in Table 5, the results on the PKUData dataset show that the accuracy of the proposed YOLOv5-PDLPR was the highest, regardless of whether Chinese characters were considered or not. When Chinese characters were considered, the accuracy of the proposed YOLOv5-PDLPR achieved 95.5%, which was 3.4% higher than the second-best algorithm. When Chinese characters were not considered, the accuracy of the proposed YOLOv5-PDLPR achieved 95.7%, which was 3.5% higher than the second-best algorithm. For the samples in the PKUData dataset, the images were captured at different moments in each day, and thus with different light levels. The proposed YOLOv5-PDLPR has not been trained on this dataset; however, it also has a very well-tested accuracy, indicating that the model is robust.

The experimental results show that the algorithm proposed in this paper has high accuracy for character recognition, can accurately recognize license plate characters even under poor lighting conditions, and has reliable generalization ability and high robustness.

### 5.3. Experiments on the AOLP Dataset

On the AOLP dataset, the experimental results of the proposed framework YOLOv5-PDLPR were compared with other detection and recognition algorithms that have performed experiments on this dataset in recent years. Two cases were considered for the localization of the position of the license plate during recognition. The first was to use the detection results of the detection network to determine the license plate position (denoted as “Box”). The second was to use the actual license plate position (denoted as “GT”). The experimental results are shown in Table 6 and Table 7. Using GT for license plate recognition can avoid errors caused by license plate positioning offset, and experimental results can more intuitively reflect the performance of the license plate recognition network. For fair comparison, we did not use any synthetic data during model training and only used rotation, translation, and scaling to extend the training dataset.

The results in Table 6 show that when using the detection results of YOLOv5 to provide license plate locations, the accuracy of the model proposed in this paper tested on AOLP-AC, AOLP-LE, and AOLP-RP was 98.5%, 99.1%, and 96.1%, respectively, which was 1.2%, 0.8%, and 1.2% higher than the second-best algorithm, respectively. The results in Table 7 show that when real license plate locations were used, the accuracies of the model proposed in this paper were 99.6%, 99.9%, and 99.8% for AOLP-AC, AOLP-LE, and AOLP-RP, respectively, which were 0.3%, 1.2%, and 5% higher than the second-best algorithm, respectively. The subset of AOLP-RP consisted mainly of rotated license plates, and the proposed method in this paper achieved the largest performance improvement in these plates. This result demonstrates that the method is effective at recognizing irregular license plates.

## 6. Ablation Study

In this section, we conducted a series of experiments to evaluate the impact of the IGFE, the parallel decoder, the number of decoding units in the parallel decoder, and the number of heads in Multi-Head Attention on the recognition accuracy. Without using a synthetic dataset, the training dataset in the experiment was half of CCPD-Base and the validation dataset was the other half. The test datasets consisted of the three sub-datasets CCPD-DB, CCPD-Tilt, and CCPD-Challenge, as these three sub-datasets best represent the impact of natural scenes such as light intensity, plate tilt, and plate blur on the performance of the license plate recognition network. The batch size for the test was set to 5.

With all other conditions being the same, using ResNet-18 [63] as the reference model, the experiments were conducted with only IGFE as the backbone as well as after adding the Focus Structure and the ConvDownSampling structure to IGFE, and the results are shown in Table 8. A “√” in the table indicates that the network used in the experiment contains the structure corresponding to this column, whereas a “×” indicates that the network used in the experiment does not contain the structure corresponding to this column.

The second and third rows in Table 8 show that by only retaining ConvDownSampling in IGFE to replace the pooling downsampling operation, the overall accuracy can be improved by 0.2 percentage points. This demonstrates that replacing the network’s pooling operation with the convolution operation can reduce the loss of features during the downsampling process and increase the ratio of correct model identification.

The second and fourth rows in Table 8 show that retaining only the Focus Structure in the IGFE improved the average accuracy by 0.4 percentage points and improved the accuracy by 0.7 percentage points on the CCPD-Challenge sub-dataset. This shows that using the Focus Structure in the network can reduce feature loss during downsampling and improve the correct rate of model recognition.

The third and fourth rows in Table 8 show that retaining only the Focus Structure in the IGFE can improve the overall accuracy by 0.2 percentage points compared to retaining only the ConvDownSampling structure in the IGFE. This is because when using the Focus Structure instead of the ConvDownSampling structure, there is no loss of feature information at any point. This makes the increase in precision attributable to the Focus Structure more apparent in the experiment results.

The second and fifth rows in Table 8 show that keeping both the Focus Structure and the ConvDownSampling structure in the IGFE can improve the overall accuracy by 1.7 percentage points; in particular, on the CCPD-Challenge sub-dataset, the accuracy was improved by 4.1 percentage points. This demonstrates that when the Focus Structure and the ConvDownSampling structure are used together, less feature information is lost during feature extraction than when the two structures are used separately. As a result, recognition accuracy is improved a lot. In addition, the first and fifth rows of Table 8 show that the accuracy of license plate recognition using IGFE was higher than that using ResNet-18. This means that the features extracted by IGFE are more complete than those extracted by ResNet-18.

To investigate the effect of using different decoders in our proposed model YOLOv5-PDLPR on license plate recognition accuracy, license plate recognition experiments were conducted on CCPD-DB, CCPD-Tilt, and CCPD-Challenge using LSTM, BiLSTM, Linear, and Parallel Decoder as decoders of the model, with all other conditions being the same, and the experimental results are shown in Table 9.

As can be seen from Table 9, when using the parallel decoder in the proposed model YOLOv5-PDLPR is more accurate than using LSTM, BiLSTM, and Linear as decoders for license plate recognition. The two decoders, LSTM and BiLSTM, as variants of RNN, have better accuracy for license plate recognition using them in model YOLOv5-PDLPR than that of Linear, the most basic fully connected layer decoder. This indicates that the parallel decoder is able to extract the global semantics of the images more adequately than the RNN under the conditions of light intensity, plate tilt and plate blurring, and the parallel decoder has higher accuracy compared with the traditional RNN decoder.

The number of heads in Multi-Head Attention submodule of the Encode module is another factor that impacts recognition performance of the proposed model YOLOv5-PDLPR. To evaluate the effect of changing the number of attention heads on the license plate recognition accuracy of the proposed model YOLOv5-PDLPR, experiments were conducted on CCPD-DB, CCPD-Tilt, and CCPD-Challenge by changing only the number of attention heads while keeping the number of decoder blocks as three and all other conditions the same, and the results are shown in Table 10.

As can be seen from Table 10, when the number of attention heads was less than or equal to eight, the license plate recognition accuracy increases with the increase in the number of attention heads on each dataset. However, when the number of attention heads exceeds eight, the license plate recognition accuracy begins to decline. This indicates that increasing the number of attention heads can improve the recognition rate; however, there is an upper limit of eight attention heads. Therefore, the number of attention heads was finally set to eight in this paper.

The recognition performance is also affected by the number of decoding units. In this section, while keeping the number of heads in Multi-Head Attention as eight and other conditions the same, the experiments were conducted on CCPD-DB, CCPD-Tilt and CCPD-Challenge by changing the number of decoder blocks, and the results are shown in Table 11.

The experimental results in Table 11 show that when the number of stacked decoding units is less than or equal to 3, the recognition accuracy of the license plate recognition model increases as the number of decoding units increases. However, when the number of decoder blocks stacked exceeds 3, the recognition effect begins to diminish. Adding more decoding units also deepens and complicates the network, which requires more calculated costs and makes training more difficult. Therefore, the number of decoder blocks was finally set to three in this paper.

## 7. Conclusions

This paper proposed a YOLOv5-PDLPR algorithm for resolving the problem of license plate detection and recognition in natural scenes under complex conditions. Compared with traditional feature extraction methods, this method included a feature extractor that can obtain global feature information, which can be used to obtain rich semantic information. Meanwhile, the advantage of multi-headed attention was fully utilized, which makes the license plate pictures accurately recognized without auxiliary correction, showing excellent performance in natural scenes. The model does not involve the RNN, so it can be inferred in parallel, which improves the recognition efficiency significantly compared with other methods. Furthermore, the experiments on the CCPD dataset achieved an average accuracy of 99.4% and recognition speed of 159.8 FPS. However, due to the limited training data set, this method can recognize fewer types of license plates and had a low recognition rate for Chinese characters other than “皖” in license plates. Therefore, in the future, the accuracy of license plate recognition can be enhanced by collecting more license plate data with a balanced distribution of Chinese characters.

## Figures and Tables

**Figure 1 sensors-24-02791-f001:**
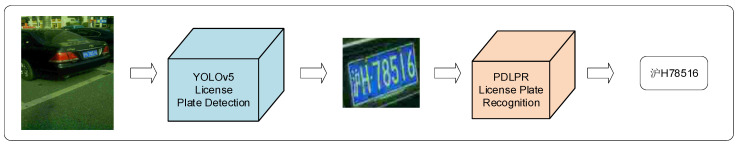
The overall process of the proposed YOLOv5-PDLPR.

**Figure 2 sensors-24-02791-f002:**
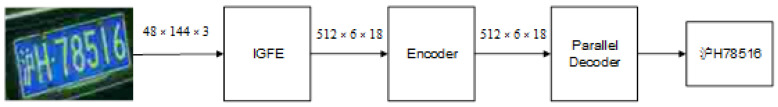
The overall framework of the license plate recognition algorithm.

**Figure 3 sensors-24-02791-f003:**
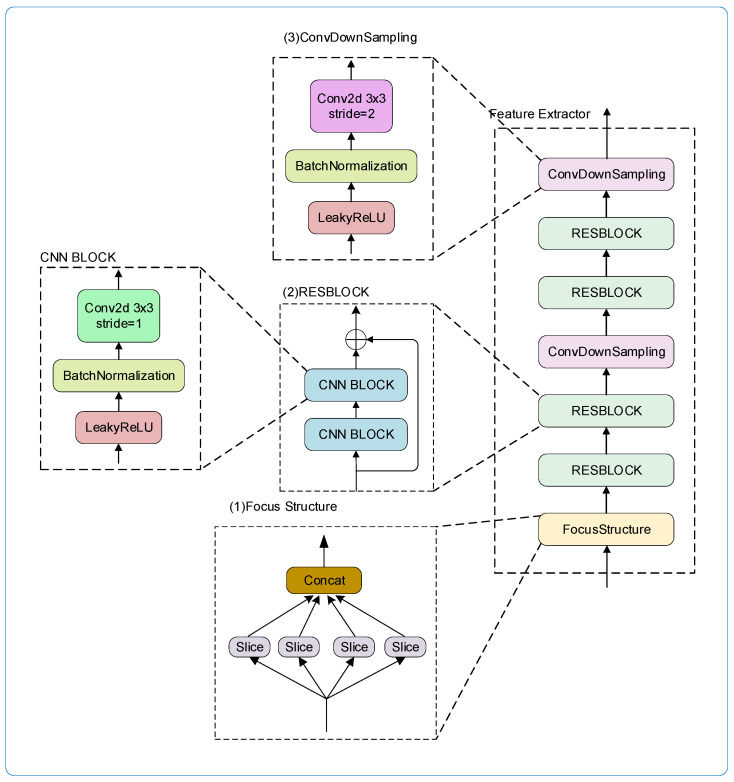
The network architecture of the IGFE.

**Figure 4 sensors-24-02791-f004:**
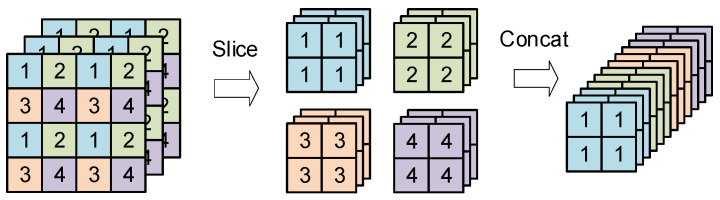
The slicing process of Focus Structure module.

**Figure 5 sensors-24-02791-f005:**
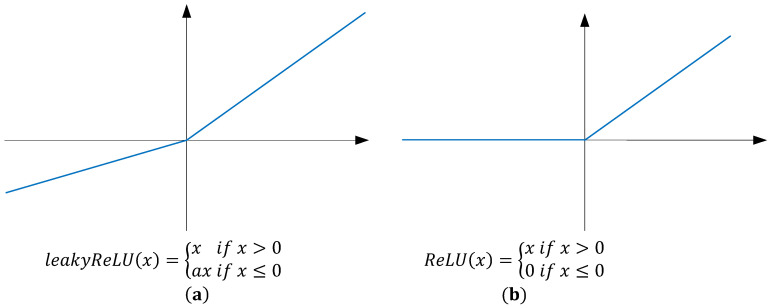
The structure of the activation functions leakyRelu (**a**) and Relu (**b**).

**Figure 6 sensors-24-02791-f006:**
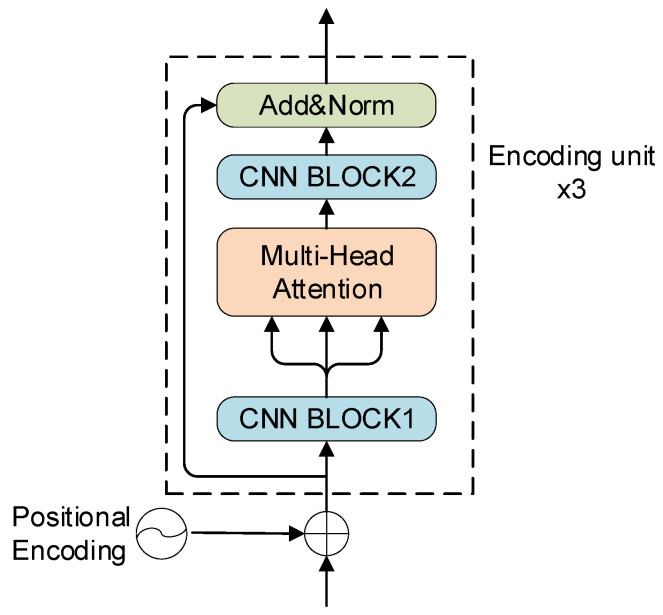
The structure of the encoder module.

**Figure 7 sensors-24-02791-f007:**
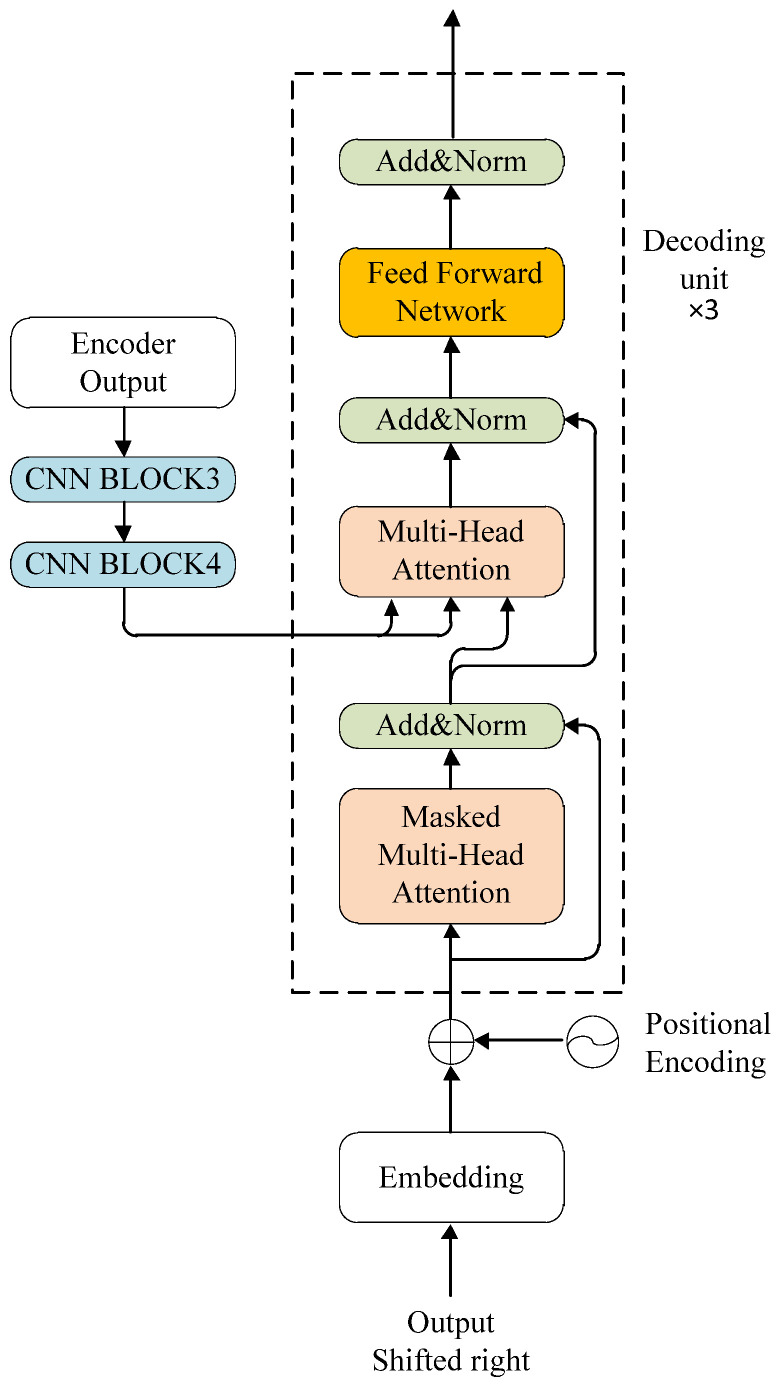
The structure of the parallel decoder network.

**Figure 8 sensors-24-02791-f008:**
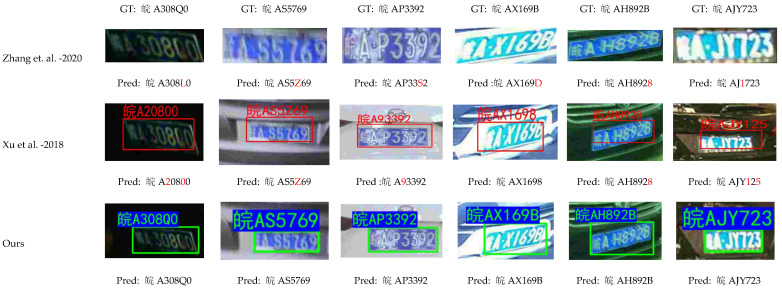
The detection and recognition results of three license plate detection recognition algorithms. If a character is displayed in red font, it means that the corresponding method does not recognize the character correctly, refs. [24,25].

**Figure 9 sensors-24-02791-f009:**
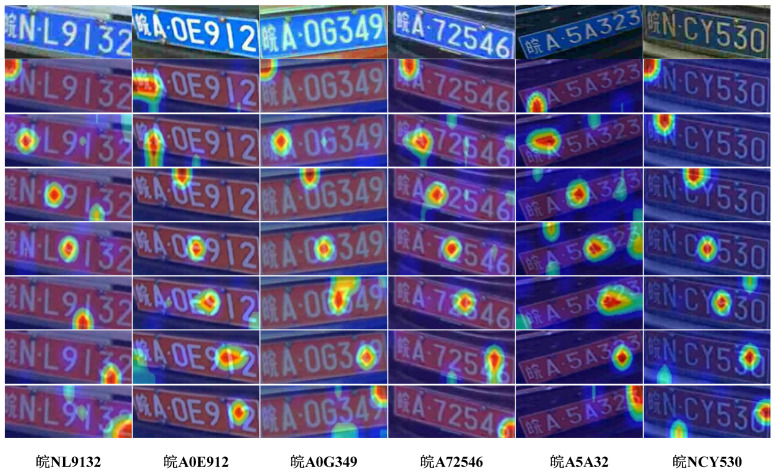
License plate recognition heat maps.

**Table 1 sensors-24-02791-t001:** The general information of the license plate datasets.

Datasets Information	CCPD	PKUData	CLPD	AOLP
Year	2018	2016	2019	2012
Number of images	283k	2253	1200	2049
Chinese province codes	29	23	31	0
Sequence length	7	7	7~8	6
Image size	720 × 1160	1082 × 727	220 × 165~4596 × 2388	640 × 480
LP colors	blue	blue + yellow	blue + yellow + green + white	white

**Table 2 sensors-24-02791-t002:** Description of each sub-dataset in the CCPD dataset [25].

Sub-Dataset	Description	Image Number
CCPD-Base	Ordinary license plate picture	200 k
CCPD-FN	License plate is relatively close or far from the camera’s shooting position	20 k
CCPD-DB	Brighter, darker or unevenly lit license plate areas	20 k
CCPD-Rotate	License plate tilted 20 to 50 degrees horizontally, −10 to 10 degrees vertically	10 k
CCPD-Tilt	License plate tilted 15 to 45 degrees horizontally and 15 to 45 degrees vertically	10 k
CCPD-Weather	License plate photographed in rain, snow and fog	10 k
CCPD-Challenge	The more challenging pictures in the plate detection recognition task	10 k
CCPD-Blur	Blurred plate images due to camera lens shake	5 k
CCPD-NP	Picture of a new car without plates fitted	5 k

**Table 3 sensors-24-02791-t003:** Comparison of the results of different license plate detection methods on the CCPD dataset. Labeling the best performance in bold and the second best performance with underlining.

Method	Overall Accuracy	Base (100 k)	DB (20 k)	FN (20 k)	Rotate (10 k)	Tilt (10 k)	Weather (10 k)	Challenge (10 k)	Speed (FPS)
Faster RCNN [9]	92.9	98.1	92.1	83.7	91.8	89.4	81.8	83.9	17.6
YOL09000 [12]	93.1	98.8	89.6	77.3	93.3	91.8	84.2	88.6	43.9
SSD300 [42]	94.4	99.1	89.2	84.7	95.6	94.9	83.4	93.1	40.7
TE2E [41]	94.2	98.5	91.7	83.8	95.1	94.5	83.6	93.1	3.2
RPnet [25]	94.5	**99.3**	89.5	85.3	94.7	93.2	84.1	**92.8**	85.5
YOLOv5	**96.7**	97.2	**97.7**	**92.9**	**98.9**	**98.9**	**99.0**	90.6	**218.3**

**Table 4 sensors-24-02791-t004:** Comparison of results of different license plate recognition methods on the CCPD dataset. Labeling the best performance in bold and the second best performance with underlining.

Method	Overall Accuracy	Base (100 k)	DB (20 k)	FN (20 k)	Rotate (10 k)	Tilt (10 k)	Weather (10 k)	Challenge (10 k)	Speed (FPS)
Ren et al., 2015 [9]	92.8	97.2	94.4	90.9	82.9	87.3	85.5	76.3	17.4
Liu et al., 2016 [42]	95.2	98.3	96.6	95.9	88.4	91.5	87.3	83.8	39.1
Joseph et al., 2016 [12]	93.7	98.1	96.0	88.2	84.5	88.5	87.0	80.5	42.0
Li et al., 2017 [41]	94.4	97.8	94.8	94.5	87.9	92.1	86.8	81.2	3.2
Zherzdev et al., 2018 [17]	93.0	97.8	92.2	91.9	79.4	85.8	92.0	69.8	56.2
Xu et al., 2018 [25]	95.5	98.5	96.9	94.3	90.8	92.5	87.9	85.1	85.5
Zhang et al., 2019 [17,61]	93.0	99.1	96.3	97.3	95.1	96.4	97.1	83.2	6.5
Luo et al., 2019 [52]	98.3	99.5	98.1	98.6	98.1	98.6	97.6	86.5	54.9
Wang et al., 2020 [53]	96.6	98.9	96.1	96.4	91.9	93.7	95.4	83.1	51.8
Zou et al., 2020 [23]	97.8	99.3	98.5	98.6	92.5	96.4	99.3	86.6	-
Zhang et al., 2020 [24]	98.5	99.6	98.8	98.8	96.4	97.6	98.5	88.9	40.2
Zhang et al., 2020 [24]	98.9	99.8	99.2	99.1	98.1	98.8	98.6	89.7	40.2
(SYNTHETIC DATA)									
Qin et al., 2021 [26]	97.2	99.3	92.9	93.2	97.9	95.5	98.8	92.4	36.0
(ResNet-18)									
Qin et al., 2021 [26]	97.6	99.5	93.3	93.7	98.2	95.9	98.9	92.9	26.0
(ResNet-50)									
Fan et al., 2022 [43]	98.8	99.7	99.1	99.0	99.1	99.3	98.5	88.0	11.7
Fan et al., 2022 [43]	99.0	99.8	99.2	99.2	**99.6**	**99.6**	98.5	88.8	26.0
(SYNTHETIC DATA)									
YOLOv5-PDLPR (Ours)	**99.4**	**99.9**	**99.5**	**99.5**	99.5	99.3	**99.4**	**94.1**	**159.8**

**Table 5 sensors-24-02791-t005:** Comparison of results of different license plate recognition methods on the CLPD and PKUData datasets. Labeling the best performance in bold and the second best performance with underlining.

Method	CLPD		PKUData	
	ACC	ACC (Without Chinese Characters)	ACC	ACC (Without Chinese Characters)
Xu et al., 2017 [25]	66.5	78.9	77.6	78.4
Zhang et al., 2020 [24]	76.8	87.6	88.2	90.5
Fan et al., 2022 [43]	55.8	79.3	81.6	81.8
Fan et al., 2022 [43] (SYNTHETIC DATA)	**82.4**	88.5	92.4	92.5
YOLOv5-PDLPR	80.3	**93.1**	**95.5**	**95.7**

**Table 6 sensors-24-02791-t006:** Comparative results of different license plate recognition methods using Box on the AOLP dataset. The best performance is labeled in bold and the second best performance with underlining.

Method	AOLP-AC	AOLP-LE	AOLP-RP
Li et al., 2017 [41]	95.3	96.6	83.7
Wu et al., 2018 [43]	96.6	97.8	91.0
Zhang et al., 2020 [24]	97.3	98.3	91.9
Zou et al., 2020 [23]	97.1	96.6	93.4
Zou et al., 2021 [62] (Box)	96.3	97.9	95.0
YOLOv5-PDLPR (Box)	**98.5**	**99.1**	**96.1**

**Table 7 sensors-24-02791-t007:** Comparative results of different license plate recognition methods using GT on the AOLP dataset. Labeling the best performance in bold.

Method	AOLP-AC	AOLP-LE	AOLP-RP
Zou et al., 2021 [62] (GT)	99.3	98.7	95.1
YOLOv5-PDLPR (GT)	**99.6**	**99.9**	**99.8**

**Table 8 sensors-24-02791-t008:** The influence of different module structures in the backbone network on the accuracy of license plate recognition. Labeling the best performance in bold.

Module Backbone	Focus Structure	ConvDownSampling	Accuracy			
			DB	Tilt	Challenge	Overall Accuracy
ResNet-18	-	-	99.0	99.3	93.3	97.7
IGFE (our)	×	×	98.8	98.3	90.3	96.6
	×	√	98.9	98.6	90.7	96.8
	√	×	99.1	98.8	91.0	97.0
	√	√	**99.5**	**99.7**	**94.4**	**98.3**

**Table 9 sensors-24-02791-t009:** The influence of different decoder on the accuracy of license plate recognition. Labeling the best performance in bold.

Decoder	Accuracy		
	CCPD-DB	CCPD-Tilt	CCPD-Challenge
LSTM	97.9	97.7	87.8
BiLSTM	96.2	95.2	80.6
Linear	90.3	81.9	70.1
Parallel Decoder	**99.5**	**99.7**	**94.4**

**Table 10 sensors-24-02791-t010:** The influence of different number of attention heads on the accuracy of license plate recognition. Labeling the best performance in bold.

Head Number	Accuracy		
	CCPD-DB	CCPD-Tilt	CCPD-Challenge
1	99.2	99.4	93.4
4	99.4	99.5	93.4
8	**99.5**	**99.7**	**94.4**
16	98.7	98.6	90.6

**Table 11 sensors-24-02791-t011:** The influence of different number of decoding units on the accuracy of license plate recognition. Labeling the best performance in bold.

Decoder Unit Number	Accuracy		
	CCPD-DB	CCPD-Tilt	CCPD-Challenge
1	97.3	94.9	84.4
2	97.8	96.4	86.4
3	**99.5**	**99.7**	**94.4**
4	99.2	99.2	91.8
5	99.0	98.7	91.0

## Data Availability

Four publicly available datasets (CCPD [25], PKUData [34], CLPD [24], and AOLP [47]) are used to validate the models.

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
