# Peer review of "A Real-Time License Plate Detection and Recognition Model in Unconstrained Scenarios"

_sensors, 2024, doi:10.3390/s24092791_

Round 1
Reviewer 1 Report
Comments and Suggestions for Authors
the manuscript has interesting topic and results. The results have presented systematically and been comparing with other results. However some notes in the introduction should be considered to meet the journal standard.
1. paragraph 5 line 60 to 71, I suggest to add some references, hence we could understood better the novelty
2. paragraph 6 line 72, "...Recently........" there is no supporting references. I suggest add references, then we could get the point.
3. Since the paper is not thesis or final year report, I suggest the authors modify the line 89 to 101
Author Response
Response to Reviewer 1’s Comments
Thanks very much for taking your time to review this manuscript. I really appreciate all your comments and suggestions! Please find my itemized responses in below and my revisions/corrections in the re-submitted files.
Comment 1: paragraph 5 line 60 to 71, I suggest to add some references, hence we could understood better the novelty
Response 1: Thank you for your valuable suggestion. Taking into account the comments of reviewer 1 and reviewer 3, we have rewritten that paragraph which you pointed out, and cited the relevant references. The following is what we revised.
Despite considerable progress in LPDR research, existing algorithms are mainly designed for recognition in stationary and constrained scenarios. License plate recognition in natural environments faces multiple challenges, including variations in shooting distance, tilt angle, light intensity, weather conditions, and image blurring[1, 27]. These factors increase the complexity of accurately recognizing license plates in natural scenes and pose a significant scientific challenge to researchers in the field. (Line 60-65 of the revised manuscript)
Comment 2: paragraph 6 line 72, "...Recently........" there is no supporting references. I suggest add references, then we could get the point.
Response 2: Thank you for your valuable the suggestion. We have added 2 new references which are [29], [30] in the revised paper, and cited them where indicated by the reviewer. The following is what we revised.
Recently, Transformer has made significant achievements in the field of Natural Language Processing (NLP)[28], and it is gradually being used in the field of computer vision with outstanding achievements[29-30]. (Line 66-68 of the revised manuscript)
Comment 3: Since the paper is not thesis or final year report, I suggest the authors modify the line 89 to 101
Response 3: Thanks for your valuable the suggestion. We accept your suggestion and revise it as follows.
- We propose a new license plate detection and recognition model called YOLOv5-PDLPR, which employs the YOLOv5 target detection algorithm in the license plate detection part and the newly proposed license plate recognition algorithm PDLPR. PDLPR has three main newly designed components: a multi-head attention mechanism for accurately recognizing individual characters, a feature extraction network for improving the integrity of the global feature extraction network, and a state-of-the-art parallel decoder architecture for improving inference efficiency.
- Experimental results on the CCPD dataset [25] show that the proposed method achieves an average accuracy of 99.4% and a recognition speed of 159.8 FPS, which are better than those of the comparison algorithms. (Line 85-94 of the revised manuscript)

Reviewer 2 Report
Comments and Suggestions for Authors
This paper proposes a new real-time license plate detection and identification model, YOLOv5-PDLPR, that combines the YOLOv5 target detection method with the PDLPR algorithm for license plate recognition. To improve accuracy and speed, the PDLPR method uses a multi-head attention mechanism, a global feature extractor network, and a parallel decoder architecture. The experimental results reveal that the proposed method beats comparison algorithms in real-time recognition, with excellent efficiency and robustness in complicated scenarios. Some existing problems still be address.
In the beginning, increasing the explanation of license plate recognition technology and the necessity for real-time capabilities could help to highlight the research's background and motivation, offering a clearer rationale and significance for the subsequent experiments and algorithm recommendations.
In terms of experimental design and result analysis, it is recommended that the author provide more extensive information regarding the experimental settings and datasets, as well as explain the reasons for selecting comparison methods. Furthermore, a more thorough analysis and explanation of the experimental data would serve to highlight the suggested algorithm's novelty and viability.
In a nutshell, the manuscript contains some original ideas and experimental value, however there is still room for development in language expression and experimental design.
Comments on the Quality of English LanguageThe lengthy sentences and complex phrases may impair reader understanding. To improve readability, it is recommended that the author optimize the article's language arrangement and logical structure.
Author Response
Response to Reviewer 2’s Comments
Thanks very much for taking your time to review this manuscript. I really appreciate all your comments and suggestions! Please find my itemized responses in below and my revisions/corrections in the re-submitted files.
Comment 1: In the beginning, increasing the explanation of license plate recognition technology and the necessity for real-time capabilities could help to highlight the research's background and motivation, offering a clearer rationale and significance for the subsequent experiments and algorithm recommendations.
Response 1: Thank you very much for your valuable comment. Based on the reviewers' comments, we have revised several parts of the paper, and blow are what we have modified:
Accurate and fast recognition of vehicle license plates from natural scene images is a crucial and challenging task. (Line 9-10 of the revised manuscript)
The license plate of a vehicle is a crucial identifier, for which accurate and real-time recognition has a very wide range of applications, such as vehicle identification, intelligent toll collection, vehicle density statistics, access control management, intelligent driving, traffic control, and traffic scene understanding. In recent years, license plate recognition has received broad attention from a wide range of researchers [1, 2]. (Line 25-29 of the revised manuscript)
In this paper, inspired by the above works and in order to be able to recognize license plates accurately and in real time, Transformer is applied to the license plate recognition task and a novel license plate detection and recognition model named YOLOv5-PDLPR is proposed. (Line 69-72 of the revised manuscript)
Comment 2: In terms of experimental design and result analysis, it is recommended that the author provide more extensive information regarding the experimental settings and datasets, as well as explain the reasons for selecting comparison methods. Furthermore, a more thorough analysis and explanation of the experimental data would serve to highlight the suggested algorithm's novelty and viability.
Response 2: Thank you very much for your valuable comment.
- We have added an introduction to the experimental server as follows.
The experiments were conducted on a server Intel(R) Xeon(R) Silver 4210 CPU @ 2.20GHz, with a GeForce RTX 2080Ti GPU and the operating system Ubuntu 18.04.6, using the neural network framework PyTorch. (Line 468-470 of the revised manuscript)
- Regarding the datasets, We politely replied that we had some differences of opinion. we believe that they have been described in detail in Section 4.1, and in addition these are publicly available datasets that have been labeled with relevant references.
- Regarding analysis and explanation of the experimental data, we make the following modifications to the paper in Sections 5.2 & 5.3:
After training the models on the CCPD-Base dataset, the models were evaluated for recognition using the CLPD and PKUData datasets, and two cases of recognition are considered: one is to recognize all characters in the license plate (including Chinese characters), and the other is to recognize only non-Chinese characters in the license plate, and is compared with the license plate recognition methods that have done the same experiments in recent years, and the experimental results are shown in Table 5. (Line 605-607 of the revised manuscript)
On the AOLP dataset, the experimental results of the proposed framework YOLOv5-PDLPR are compared with other detection and recognition algorithms that have done experiments on this dataset in recent years. (Line 639-640 of the revised manuscript)
- Regarding the analysis and interpretation of the experimental data, We politely replied that we had some differences of opinion. We believe that reasonable explanations and analyses have been done in each experiment.
Comment 3: In a nutshell, the manuscript contains some original ideas and experimental value, however there is still room for development in language expression and experimental design.
Comment 4: The lengthy sentences and complex phrases may impair reader understanding. To improve readability, it is recommended that the author optimize the article's language arrangement and logical structure.
Response 3 & 4: Thank you very much for your valuable comment. We apologize for the poor language of our manuscript. We worked on the manuscript for a long time and the repeated addition and removal of sentences and sections obviously led to poor readability. We have now worked on both language and readability and have also involved native English speakers for language corrections. We really hope that the flow and language level have been substantially improved

Reviewer 3 Report
Comments and Suggestions for Authors
The paper has scientific novelty as it proposes a new algorithm based on the YOLOv5 neural network. The method is proposed for the recognition of car license plates. However, there are a number of remarks to the paper:
1) The authors emphasize on fast inference, they should also indicate works such as (10.1134/S1054661822030038, https://arxiv.org/pdf/1903.07414.pdf) related to optical flow in the review part.
2) Lines 60-71: rewrite by removing the ChatGPT style
3) Line 102: says 159.8 FPS performance, but does not say on which device this is obtained.
4) Car license plates can go at a slant. Is a model with oriented bounding rectangles used?
5) In Figure 3, there are no pooling layers in the architecture. What is the rationale for this?
6) Line 346: figure caption got on another page - correct (should be on the same page as the figure).
7) Table 3 - Last column - it would be better to call the figure "Model Performance" rather than Inference Time, since it is FPS, not time. It is not clear on which computing device such high processing speeds are obtained.
8) Similarly for Table 4.
9) Figure 9. Transfer the sub-drawing caption to the same page as the figure.
10) Table 5. typo "OLOv5-PDLPR" (should be YOLO)
Author Response
Response to Reviewer 3’s Comments
Thanks very much for taking your time to review this manuscript. I really appreciate all your comments and suggestions! Please find my itemized responses in below and my revisions/corrections in the re-submitted files.
Comment 1: The authors emphasize on fast inference, they should also indicate works such as (10.1134/S1054661822030038, https://arxiv.org/pdf/1903.07414.pdf) related to optical flow in the review part.
Response 1: Thank you very much for your valuable suggestion. We have carefully read the literature you listed, these literatures can really help us to have a deeper and broader understanding of Optical Flow CNN algorithms, thus adopted your suggestion, and have added 2 new references which are [44], [45] in the revised manuscript, and cited them at Line 158 of the revised manuscript.
Comment 2: Lines 60-71: rewrite by removing the ChatGPT style
Response 2: Thank you very much for your valuable comment. Taking into account the comments of reviewer 1 and reviewer 3, we have rewritten that paragraph which you pointed out, and cited the relevant references. The following is what we revised.
Despite considerable progress in LPDR research, existing algorithms are mainly designed for recognition in stationary and constrained scenarios. License plate recognition in natural environments faces multiple challenges, including variations in shooting distance, tilt angle, light intensity, weather conditions, and image blurring[1, 27]. These factors increase the complexity of accurately recognizing license plates in natural scenes and pose a significant scientific challenge to researchers in the field. (Line 60-65 of the revised manuscript)
Comment 3: Line 102: says 159.8 FPS performance, but does not say on which device this is obtained.
Response 3: Thank you for the suggestion. We have added a description of the experimental device in Section 4.2. Below is what we revised in section 4.2.
The experiments were conducted on a server Intel(R) Xeon(R) Silver 4210 CPU @ 2.20GHz, with a GeForce RTX 2080Ti GPU and the operating system Ubuntu 18.04.6, using the neural network framework PyTorch. (Line 468-470 of the revised manuscript)
Comment 4: Car license plates can go at a slant. Is a model with oriented bounding rectangles used?
Response 4: Thank you very much for your valuable comment. There is no model with oriented bounding rectangles in the proposed model YOLOv5-PDLPR. In YOLOv5-PDLPR, in the recognition phase, it can find the location of the license plate by itself, regardless of whether the license plate is tilted or not, because it is a deep learning based model that has been trained. Of course, in the training phase, the location of the license plate and the information of the license plate need to be labeled.
Comment 5: In Figure 3, there are no pooling layers in the architecture. What is the rationale for this?
Response 5: Thank you very much for your valuable comment. It remains to be clarified that in “section 3.2.1 (3) ConvDownSampling module” there is an introduction to use convolution operation instead of pooling operation for downsampling, because this way more feature information can be retained during downsampling, which improves the accuracy of license plate recognition.
Comment 6: Line 346: figure caption got on another page - correct (should be on the same page as the figure).
Response 6: Thanks for pointing this out. We have revised it in the revised version of the paper.
Comment 7: Table 3 - Last column - it would be better to call the figure "Model Performance" rather than Inference Time, since it is FPS, not time. It is not clear on which computing device such high processing speeds are obtained.
Response 7: Thank you very much for your valuable comment. We have revised it to “Speed(FPS)”, and the following is the revised version. The computing device is introduced in the response to comment 3.
Table 3. Comparison of the results of different license plate detection methods on the CCPD dataset. Labeling the best performance in bold.
|
Method |
Overall |
Base |
DB |
FN |
Rotate |
Tilt |
Weather (10k) |
Challenge |
Speed (FPS) |
|
Faster RCNN [9] |
92.9 |
98.1 |
92.1 |
83.7 |
91.8 |
89.4 |
81.8 |
83.9 |
17.6 |
|
YOL09000 [12] |
93.1 |
98.8 |
89.6 |
77.3 |
93.3 |
91.8 |
84.2 |
88.6 |
43.9 |
|
SSD300 [38] |
94.4 |
99.1 |
89.2 |
84.7 |
95.6 |
94.9 |
83.4 |
93.1 |
40.7 |
|
TE2E [37] |
94.2 |
98.5 |
91.7 |
83.8 |
95.1 |
94.5 |
83.6 |
93.1 |
3.2 |
|
RPnet [25] |
94.5 |
99.3 |
89.5 |
85.3 |
94.7 |
93.2 |
84.1 |
92.8 |
85.5 |
|
YOLOv5 |
96.7 |
97.2 |
97.7 |
92.9 |
98.9 |
98.9 |
99.0 |
90.6 |
218.3 |
Comment 8: Similarly for Table 4.
Response 8: Thank you very much for your valuable comment. We have revised it to “Speed(FPS)”, and the following is the revised version. The computing device is introduced in the response to comment 3.
Table 4. Comparison of results of different license plate recognition methods on the CCPD dataset. Labeling the best performance in bold.
|
Method |
Overall |
Base |
DB |
FN |
Rotate |
Tilt |
Weather |
Challenge |
Speed(FPS) |
|
Ren et al. -2015 [9] |
92.8 |
97.2 |
94.4 |
90.9 |
82.9 |
87.3 |
85.5 |
76.3 |
17.4 |
|
Liu et al. -2016 [38] |
95.2 |
98.3 |
96.6 |
95.9 |
88.4 |
91.5 |
87.3 |
83.8 |
39.1 |
|
Joseph et al. -2016 [12] |
93.7 |
98.1 |
96.0 |
88.2 |
84.5 |
88.5 |
87.0 |
80.5 |
42.0 |
|
Li et al. -2017 [37] |
94.4 |
97.8 |
94.8 |
94.5 |
87.9 |
92.1 |
86.8 |
81.2 |
3.2 |
|
Zherzdev et al. -2018 [17] |
93.0 |
97.8 |
92.2 |
91.9 |
79.4 |
85.8 |
92.0 |
69.8 |
56.2 |
|
Xu et al. -2018 [25] |
95.5 |
98.5 |
96.9 |
94.3 |
90.8 |
92.5 |
87.9 |
85.1 |
85.5 |
|
Zhang et al. -2019 [40] |
93.0 |
99.1 |
96.3 |
97.3 |
95.1 |
96.4 |
97.1 |
83.2 |
6.5 |
|
Luo et al. -2019 [48] |
98.3 |
99.5 |
98.1 |
98.6 |
98.1 |
98.6 |
97.6 |
86.5 |
54.9 |
|
Wang et al. -2020 [49] |
96.6 |
98.9 |
96.1 |
96.4 |
91.9 |
93.7 |
95.4 |
83.1 |
51.8 |
|
Zou et al. -2020 [23] |
97.8 |
99.3 |
98.5 |
98.6 |
92.5 |
96.4 |
99.3 |
86.6 |
- |
|
Zhang et al. -2020 [24] |
98.5 |
99.6 |
98.8 |
98.8 |
96.4 |
97.6 |
98.5 |
88.9 |
40.2 |
|
Zhang et al. -2020 [24] |
98.9 |
99.8 |
99.2 |
99.1 |
98.1 |
98.8 |
98.6 |
89.7 |
40.2 |
|
(SYNTHETIC DATA) |
|||||||||
|
Qin et al. -2021 [26] |
97.2 |
99.3 |
92.9 |
93.2 |
97.9 |
95.5 |
98.8 |
92.4 |
36.0 |
|
(ResNet-18) |
|||||||||
|
Qin et al. -2021 [26] |
97.6 |
99.5 |
93.3 |
93.7 |
98.2 |
95.9 |
98.9 |
92.9 |
26.0 |
|
(ResNet-50) |
|||||||||
|
Fan et al. -2022 [39] |
98.8 |
99.7 |
99.1 |
99.0 |
99.1 |
99.3 |
98.5 |
88.0 |
11.7 |
|
Fan et al. -2022 [39] |
99.0 |
99.8 |
99.2 |
99.2 |
99.6 |
99.6 |
98.5 |
88.8 |
26.0 |
|
(SYNTHETIC DATA) |
|||||||||
|
YOLOv5-PDLPR(Ours) |
99.4 |
99.9 |
99.5 |
99.5 |
99.5 |
99.3 |
99.4 |
94.1 |
159.8 |
Comment 9: Figure 9. Transfer the sub-drawing caption to the same page as the figure.
Response 9: Thanks for pointing this out. We have revised it in the revised version of the paper.
Comment 10: Table 5. typo "OLOv5-PDLPR" (should be YOLO)
Response 10: Thanks for pointing this out. It was indeed a clerical error, it was our mistake and we have corrected it (in last line of Table 5 of the revised manuscript), and the following is the revised version..
Table 5. Comparison of results of different license plate recognition methods on the CLPD and PKUData datasets. Labeling the best performance in bold.
|
Method |
CLPD |
PKUData |
||
|
ACC |
ACC(Without |
ACC |
ACC(Without |
|
|
Xu et al. 2017 [25] |
66.5 |
78.9 |
77.6 |
78.4 |
|
Zhang et al.-2020 [24] |
76.8 |
87.6 |
88.2 |
90.5 |
|
Fan et al. -2022 [39] |
55.8 |
79.3 |
81.6 |
81.8 |
|
Fan et al. -2022 [39] (SYNTHETIC DATA) |
82.4 |
88.5 |
92.4 |
92.5 |
|
YOLOv5-PDLPR |
80.3 |
93.1 |
95.5 |
95.7 |
